# Synergy Degree Evaluation of Container Multimodal Transport System

**Xiaoping Fang \*, Zhang Ji, Zhiya Chen, Weiya Chen \*****, Chao Cao** **and Jinrong Gan**

School of Traffic and Transportation Engineering, Rail Data Research and Application Key Laboratory of Hunan Province, Central South University, Changsha 410075, China; jizhang970817@csu.edu.cn (Z.J.); chzy@csu.edu.cn (Z.C.); caochao@csu.edu.cn (C.C.); ganjinrong@csu.edu.cn (J.G.)
\* Correspondence: fangxp@csu.edu.cn (X.F.); wychen@csu.edu.cn (W.C.)

**Abstract:** Logistics activities are an important source of energy consumption and environmental issues. Research conclusions and practical experience show that promoting the development of container multimodal transport is an effective way to reduce the level of carbon footprint. The key to influencing the development of container multimodal transport lies in the cooperation of all participants and links (modes of transport, transport businesses). Evaluating the synergy degree is a key step in this development process. This paper takes the whole process of container multimodal transportation as the research perspective, analyzes the operation process, and treats the process as a production system composed of four subsystems: facilities and equipment, organizational management, business operations, and information interactions. Through in-depth interviews and an analysis of the academic literature and policy documents, we establish a synergy degree evaluation index system and measurement model of container multimodal transport based on synergy theory and case studies. The research results are consistent with the actual situation. From 2015 to 2018, the synergy of container multimodal transport system of China's G port developed slowly, but generally moved in a more orderly direction.

**Keywords:** container multimodal transport system; grounded theory; synergy degree; order degree

## 1. Introduction

A report by the International Energy Agency shows that the transportation sector accounts for nearly a quarter of current energy-related $CO_2$ emissions. Voluntary public reductions in carbon-intensive travel behavior [1] and the development of low-carbon logistics have become an important countermeasure for sustainable development. They have been increasingly valued by government departments and enterprises and have attracted the attention of many scholars. Among them, low-carbon logistics and reduction of energy consumption and carbon footprint have become the inevitable trend of the development of the logistics industry [2]. Low-carbon logistics is conducive to energy conservation and footprint reduction for logistics companies, improving market competitiveness [3] and building a sustainable economic system with low-carbon manufacturing and consumption. Compared with single road transportation, multimodal transportation can effectively reduce transportation costs by 20% [4] and $CO_2$ footprint by 57% [5] and is an effective and sustainable transportation method. In 2016, the Paris Agreement put forward energy conservation and carbon footprint reduction targets: "By 2020, China's unit gross domestic product $CO_2$ footprint will fall by 40% to 45%; by 2030 it will fall by 60% to 65%." Multimodal container transportation can improve transportation efficiency, reduce transportation costs, and promote energy conservation and environmental protection. In 2018, the State Council of China issued the Three-Year Action Plan to Promote the Adjustment of Transport Structure (2018–2020), which proposes optimizing the structure

of cargo transportation; by 2020, the volume of port railway collection and distribution and container multimodal transport will increase significantly. Multimodal container operation is an important economic activity and plays a vital role in transportation and low-carbon development.

In actual operation, logistics companies tend to choose a single mode of transport when conditions permit. The reason is that the participants and links (modes of transport, transport businesses) are not coordinated, due to the following: (1) There is a lack of effective connection between infrastructures. For example, many ports have no dedicated railway lines, which greatly increases transportation costs. (2) There is no uniform standard for equipment. The size standards of road trucks are different from the standards of railway loading units, and the conversion is very troublesome, requiring loading and unloading, resulting in increased costs. (3) The devices are not completely intelligent. The development of special railway wagons, railway dual-use trailers, and river–sea combined carriers is lagging, and the connection efficiency is low. (4) There is a lack of multimodal transport integration service providers. Due to the lack of a large logistics network platform and effective deployment, small transportation companies are blind, and there is often repeated transportation and wasting of resources. (5) Information docking and sharing are difficult. There is no uniform information exchange standard and format between enterprises, and each enterprise is an information island. In addition, there is a lack of effective connection between multiple modes of transport, and the costs of short barges, reloading, loading and unloading, and distribution are relatively high. Zhou Zhicheng, director of the Research Office of China Federation of Logistics and Purchasing, said that combined with short-distance transportation and inconsistent equipment standards, railway transportation is no longer competitive compared to single-channel transportation. Therefore, Li Muyuan, secretary general of the China Multimodal Transport Association, believes that the success of multimodal transport depends on whether it can reduce transaction costs between enterprises and links and build a cooperative network among the main bodies to achieve a high degree of strategic coordination among all links.

Researchers and managers not only recognize that traditional single transport does not meet the needs of modern cities [6], but also recognize the nature of multimodal transportation collaborative work. However, the current research results of multimodal transport collaboration focus on optimization aspects, such as route optimization and transportation mode selection, and follow the "linear optimization" line of thought. Container multimodal transport itself is not a completely linear system. In practice, there are many interchange points and external environmental factors that make it impossible for goods to flow smoothly.

Analysis of the effects of synergies on many aspects of the supply chain shows that it helps reduce carbon footprint by reducing the bullwhip effect [7], using RFID and other IoT technologies [8] to reduce errors in transportation and the uncertainty of the supply chain [9,10]. We all recognize that multimodal transport is an important area of supply chain research. We measure the degree of collaboration of container multimodal transport based on the synergy theory, which can provide managers with future directions for management and help improve the quality of multimodal transport services so that more managers can use multimodal transport to transport goods. With the increase of the proportion of multimodal transport in the total transport volume, the carbon footprint of a certain amount of social logistics will be reduced.

Evaluation is the first step in building synergy. Only by taking the long and complex chain link activity in a system, studying the collaborative process in conjunction with the actual situation, assessing it, and discovering weaknesses can managers formulate a plan, achieve efficiency growth, and promote coordinated construction of the entire process of container multimodal transport. Therefore, evaluating the synergy degree of the system is the key and the first step toward achieving low-carbon logistics, which can give container multimodal transport a role in transportation and low-carbon development. This is the goal of this paper.

We set out to measure the synergy degree of container sea–rail transportation. We propose an evaluation index system and measurement model for measuring the synergy degree of the container multimodal transport system to study its collaborative nature. Specifically, we establish an evaluation

index system by analyzing the academic literature and policy documents, conducting qualitative research analysis based on grounded theory, and using metric models to calculate and conduct empirical analysis. The rest of this paper is as follows: The second part is a literature review of synergy theory and container multimodal transport. The third part is the content and target analysis of the system. In the fourth part, the evaluation index system for the synergy degree of the system is established. In the fifth part, the measurement model of the synergy degree of the system is given and analyzed through a case study.

## 2. Literature Review

### 2.1. Synergy Theory

Haken defines collaboration as a system's components working together to form a new structure and feature that does not exist at the individual level. When the subsystems are more interconnected, mutual forces can affect the control system's operation, and the system will present orderly characteristics on the macro level [11]. There are three necessary conditions for the system to achieve synergy and order: the system must be open, the system must have nonlinear properties, and there must be synergy between subsystems. As the system moves closer to the critical point where significant qualitative changes occur, fast and slow variables in the system will move. Fast variables guide the system to return from an unstable state to a stable state, while slow variables guide the system to move from the original stable state to an unstable state, thereby evolving to a new stable state. These slowly changing parameters are also called order parameters, which determine the macroscopic behavior of the system and characterize its degree of ordering. They guide the system to form new structures and features through competition and collaboration. The phenomenon of self-organization is that in the process of migration, fast and slow variables appear to be related and limited, thus showing a macroscopic reflection of coordinated movement. The process of system collaboration based on synergy is shown in Figure 1. The order parameter is the core concept of synergy theory, reflecting the degree of order of the new structure of the system [12].

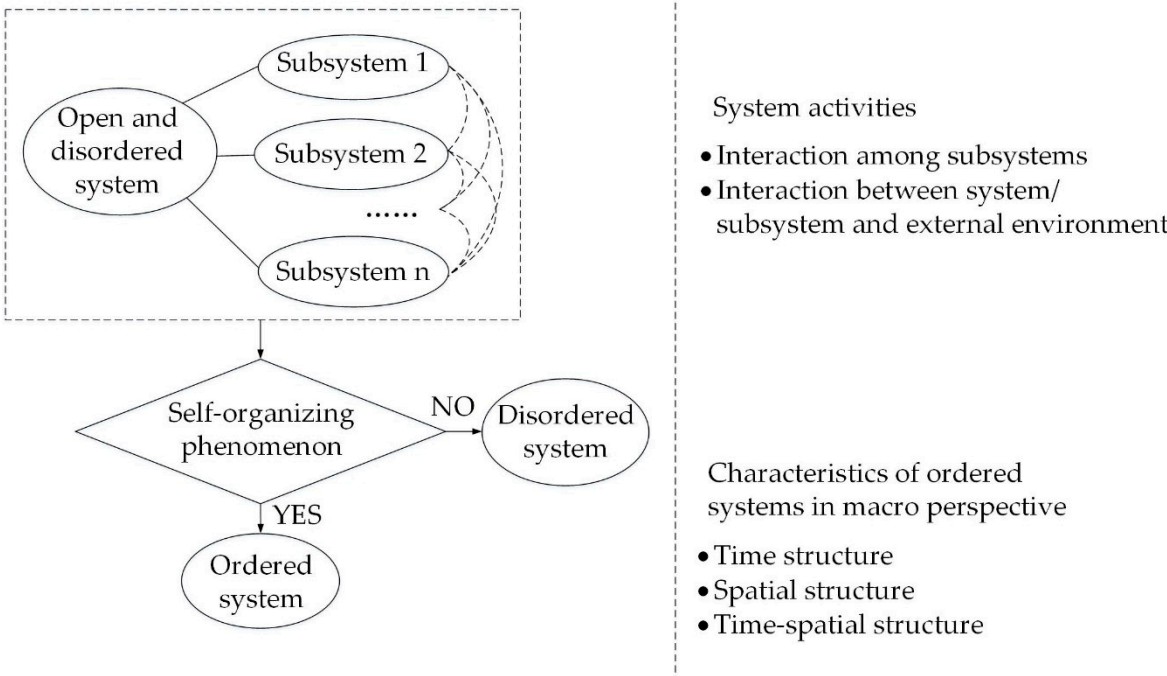

**Figure 1.** Process of system collaboration based on synergy.

Synergy theory can not only explain the relationship between the interaction and cooperation of a system, but also can judge the degree of coordinated development of internal subsystems and elements, find the elements that reduce the synergy degree, and manage the synergy between subsystems [13]. Therefore, synergy theory is widely used in the fields of economics and social sciences [14,15] and the self-organization of urban ecology [16–18]. Qiao et al. regarded the urban water system landscape as a composite ecosystem composed of environment, economy, and society, found the key factors that affect the synergy of various subsystems based on synergy theory [19]. Jiao et al. studied the synergy of sustainable urbanization in the same process [20]. Cui took the supply and demand coordinated development mechanism of a regional logistics system as the research goal, and analyzed the coordinated development mechanism of the regional logistics supply and demand coupling system [21]. Wang et al. proposed a novel approach to construct an indicator system to evaluate the development of a land–sea coordination system, with the results providing a scientific reference and decision-making basis for policy formulation and development evaluation of standard ports [22].

Synergy theory is also applicable to research on sustainable supply chain operations. Supply chain collaboration is a network association formed by two or more companies through agreements or joint organizations to achieve a certain strategic purpose [23]. As corporate collaboration increases, so does the level of supply chain performance [24]. Sandberg analyzed logistics collaboration in Swedish supply chain management with a questionnaire. The research results show that there is a direct relationship between the strength of logistics collaboration and the level of collaborative performance. Effective supply chain management can enhance the degree of logistics collaboration [25]. Huang et al., Wu, and Chiu focused on the internal collaboration of the supply chain, analyzed the importance of resources and information, and used collaborative methods to evaluate and improve operational efficiency [26,27]. Soylu, A et al., Sodenkamp et al., and Sun et al. focused on synergy in the energy supply chain management of different enterprises, supplier selection and order allocation, rail transport and trade. The synergy effect between them improves operation efficiency [28–30].

*2.2. Container Multimodal Transport*

Multimodal transport is defined as the transportation of goods by at least two different modes of transport (UNECE). It is the joint coordination of multiple modes of transport by road, sea, rail, and air. It emphasizes the integration and seamless connection between transport modes [31]. Multimodal container transport uses containers as the transport unit and organically combines different modes to form continuous and comprehensive integrated cargo transportation [32]. The purpose is to reduce the loading and unloading of cargo, reduce damage, improve transportation efficiency, and optimize the overall transportation benefits at the same time. These characteristics make container multimodal transport a complex system with multiple participants and links, involving a wide range of interacting stakeholders, decision-makers, and operational planning activities [33]. Its objectives and interactions are subject to the influence of external uncertainties. Therefore, the operation of the container multimodal transport chain requires the cooperation of all participants [34] and the integration, coordination, and seamless connection of all links. Through the above analysis, we find that (1) research on synergy in multimodal transport is scarce, and multimodal transport is essentially a type of supply chain, so we can learn from research on synergy in supply chains, and (2) the intermodal transportation system satisfies the three conditions of orderly coordination: it is open, nonlinear, and the subsystems need to cooperate with each other. Therefore, it is appropriate and reasonable to research the synergy of container multimodal transport.

Because of its length and complexity, the coordinated operation of container multimodal transport has received extensive attention and research. Many scholars have conducted research from different perspectives of managing transportation activities, such as organizational management, business activities, and the impact of external risk on support transportation decisions. From the perspective of organizational management, scholars use effective cooperation of internal freight forwarders to coordinate transportation processes [35] or use external customer needs as a breakthrough

point to establish reasonable transportation routes and methods [31] to achieve logistical coordination and minimize costs. At the same time, the service quality evaluation work is also a very reasonable direction. Through the analysis of secondary data from academic papers, government policies, and industry reports, the quality characteristics of multimodal transport services are clarified. Through the analysis of the firsthand data of in-depth interviews with multimodal transport practitioners, the service quality evaluation index that conforms to the theory and practice is established [36]. From a business connection perspective, some scholars have looked at operational management by coordinating the profits between multimodal enterprises [37] and the interests among port stakeholders [38]. Considering the perspective of external risks, some scholars have focused on the impact of risk factors, climate environment, and changes in water levels [39] on intermodal transport, and helped transport networks operate smoothly through case studies. Trans-shipment scheduling is also very important. There are some studies on the impact of container trans-shipment [40], lock and trans-shipment co-scheduling [41], and scheduling management for vessel traffic at ports [42] on intermodal operations, and reasonable methods have been proposed to improve operational efficiency.

In Sections 2.1 and 2.2, we note the following: (1) The theoretical foundation of synergy is relatively complete and is beginning to expand into the field of integrated transportation, but there is less research on measuring the synergy degree of container multimodal transport systems. (2) Research on multimodal transportation is rarely viewed from the perspective of multiparty collaboration, systems, and holistic aspects, including software and hardware, business management, and organizational operations. (3) In research on supply chain collaboration, there are more studies on internal collaboration of enterprises. However, it is less connected with customers at both ends and lacks the awareness of the whole process. (4) There is more research on the connection operation of multimodal transport from a single aspect and the lack of integration with the basic theory of collaboration. Therefore, we will look at multimodal transport from the perspective of the whole process, including software and hardware collaboration (information resources, facility resources), internal and external collaboration (between enterprise members, between enterprises and customers), orderly rotation of the operation chain, and optimization of resource allocation, to optimize the transportation structure, which plays a very important role in management decisions.

## 3. Collaborative Analysis of Container Multimodal Transport System

### 3.1. Composition of Container Multimodal Transport System

The American scholar Ackoff describes a system as a collection of two or more elements of any kind that are related to each other, and that interact and depend on each other. This is in line with the concept of synergy in the container multimodal transport supply chain. According to the actual operation of container multimodal transport, the system is regarded as a collection of software and hardware resources and coordinated internal and external operations. In the actual production and operation process, facilities and equipment as hardware resources and information as a software resource provide labor data for production activities. The interactions among enterprise members, enterprises, and customers produce the necessary organizational management and business operation process. In order to further reflect on the operating characteristics of the system and emphasize the concept of sustainable development, we define the container multimodal transport system as a production system consisting of interactive subsystems of business operations, organization and management, facilities and equipment, and information interaction, in which the coordinated interaction of the core subsystems will produce a synergistic effect of 1 + 1 > 2. The definition of system collaboration is that all members of the container multimodal transport company cooperate with each other to achieve the strategic goal of efficient transportation and sustainable development through the sharing of information and close connection of transportation methods. The structure of the container multimodal transport system is shown in Figure 2.

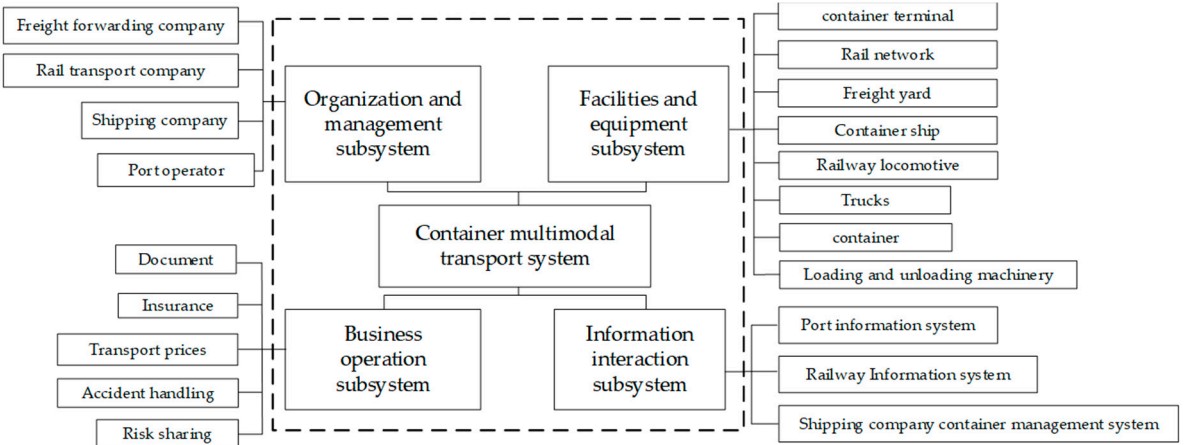

**Figure 2.** Structure of container multimodal transport system.

## 3.2. Collaborative Content Analysis of Container Multimodal Transport System

Under the premise that the nature of transported goods has been determined, the actual conditions of organizational management and facilities and equipment determine the possible transport modes of multimodal products, while business operations and information interactions provide guarantees for the smooth operation of transport organizations. Specifically: (1) the organization and management subsystem establishes a standardized organizational process and rationally allocates resources to achieve rapid response and low-carbon transportation; (2) the configuration of the facilities and equipment subsystem includes facility layout matching and equipment specialization to achieve transportation, optimize resources, and improve efficiency; (3) in the process of cargo transportation, the business operation subsystem ensures the smooth connection of business processes, including the connection of documents, customs clearance administration, claims for cargo damage, etc., which plays an important role in profit distribution; and (4) organizational management, deployment of equipment, and business affairs all require accurate information provided by the information interaction subsystem to enable member enterprises to communicate smoothly at each link and realize the simultaneous transportation of information and goods. The four subsystems work together to achieve the requirements of high transport efficiency for logistics companies, low transaction costs for cargo owners, low cargo damage and spread, and low carbon emissions. In other words, the synergy of the sequential parameters of the business operation, organizational management, facilities and equipment, and information interaction subsystems guide the evolution of the whole system to achieve overall efficiency (E), sustainable development (S), profit (P), and quality (Q) goals. The relationship between synergy and the goals of the container multimodal transport system is shown in Figure 3.

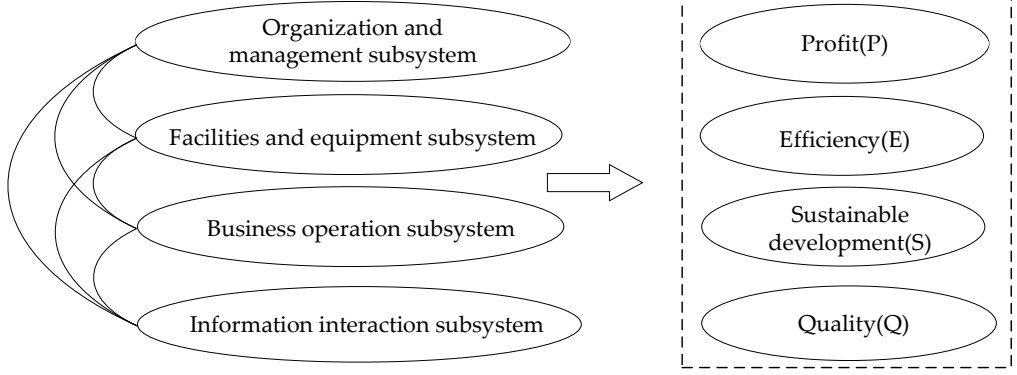

**Figure 3.** Relationship between the synergy and the goals of the container multimodal transport system.

## 4. Establishing Evaluation Indicator System for Synergy Degree of Container Multimodal Transport System

Establishing the evaluation indicator system for the synergy degree of the container multimodal transport system is the basis for testing its synergy degree. The four subsystems of the system are used as four secondary indicators, and 20 sub-elements (Figure 2) are used as starting points to determine the evaluation indicators of the synergy degree. Combined with the theoretical research and actual development of container multimodal transport, this is analyzed from three aspects: (1) selecting multimodal transport coordination and coordination evaluation indicators from relevant academic literature, (2) conducting surveys of multimodal transport service providers and customers to understand the actual needs of coordinated operations to determine indicators, and (3) analyzing and determining indicators in government-related development plans, standards, and industry association reports.

### 4.1. Selection of Indicators Based on Academic Literature Analysis

The development of multimodal transport is of great significance to low-carbon transportation and the sustainable development of society, and has attracted the attention of scholars. We used the keywords "multimodal transportation", "intermodal transportation", "combined transportation", "container", "synergy", "synergetic", "coordinate", "coordination", "evaluation", and "measurement" to conduct a combined search in the Chinese and English academic resource databases to effectively screen the indicators and standards mentioned in the literature. The following can be seen from Table 1: (1) The frequency of "environmental costs" mentioned in the English literature is twice the frequency in the Chinese literature (corresponding to "pollution emissions" in the Chinese literature). China's comprehensive transportation development has paid a relatively large environmental price, and environmental cost should be used as an important indicator. (2) The Chinese literature pays more attention to timely compensation for cargo damage, investment in information construction, and administrative efficiency of customs clearance. (3) The meanings of the indicators are somewhat different. In the Chinese literature, "cargo station capacity matching degree" focuses on descriptions of the capacity of the freight station, while the English "transport capacity" has a broader definition of capacity, including fixed equipment (line, station, port, terminal, berth, warehouse, etc.), the amount of equipment (vehicles, ships, etc.), the number of staff members, etc. (4) The frequency gap between the English literature indicator "informatization level" and the Chinese literature indicator "proportion of information construction investment" is large, and the amount of information on European and American countries' multimodal transport is relatively large, focusing on assessing the quality of information services. China's multimodal transport is in the development stage, and it is more focused on investment in informatization construction. (5) The phrases "mechanization level of facilities and equipment", "efficiency of container dispatch operations", "standardization level of electronic documents", "efficiency of customs clearance administration", and "time compensation rate for cargo loss" have never appeared in the related English literature. In order to fully consider the container multimodal transport process, we have added these indicators to the final indicator system.

**Table 1.** Indicators from English and Chinese literature.

| Subsystem | English Literature Index | Frequency | Chinese Literature Index | Frequency |
|---|---|---|---|---|
| Facilities and equipment subsystem | Construction of infrastructure | 5 | Degree of matching between equipment and facilities | 6 |
| | Transport capacity | 11 | Cargo station capacity matching degree | 3 |
| | | | Mechanization level of facilities and equipment | 13 |
| Organization and management subsystem | Multimodal capacity | 6 | Rapid multimodal capacity | 8 |
| | Efficiency of transfer operations | 11 | Efficiency of transport connection | 8 |
| | | | Efficiency of container dispatch operations | 8 |
| | Environmental costs | 13 | Pollution emissions | 6 |
| Information interaction system | Information level | 9 | Proportion of information construction investment | 22 |
| | Ability to track cargo | 3 | Quality of cargo tracking | 8 |
| business operation subsystem | Connection at both ends | 5 | Proportion of connection costs at both ends | 7 |
| | | | Standardization level of electronic documents | 9 |
| | | | Efficiency of customs clearance administration | 17 |
| | | | Time compensation rate for cargo loss | 37 |

*4.2. Selection of Indicators Based on In-Depth Interviews with Container Multimodal Transport Service Providers and Customers*

In order to know the obstacles in the entire process of container multimodal transport operation and collaboration, we conducted open and in-depth interviews with personnel at representative ports and railway freight companies, and with customers. Grounded theory uses nonprobabilistic and intentional sampling, so it has the characteristics of strong purpose and very specific analysis of small samples. Considering the perspective of the entire process of establishing indicators, we selected multimodal service providers and conducted in-depth interviews with customers in order to make the indicators fully consider the needs of every business entity. From November 2018 to February 2019, we conducted in-depth interviews with senior practitioners and cargo owners (customers) of seven intermodal suppliers in Hunan, Guangdong, and Shandong. The topics and outline of the interview are shown in Table 2, but it is not limited to the issues listed in the table.

After that, we analyzed the interview data, mined information, and established conceptual categories (indicators) based on grounded theory, as shown in Appendix A. The indicators are ranked by the number of times respondents mentioned them. It can be seen in Appendix A that the number of sample points for the intelligence degree of technical equipment, the matching degree of the railway line of the port, and the matching degree of capacity and demand were 26, 16, and 15, respectively. They have been the focus of intermodal service providers and customers because (1) the technology and equipment of China's multimodal transport have not yet achieved intelligence, (2) many ports do not have dedicated railway lines, and have added short-haul transportation links, which makes the transfer process more complicated and cumbersome, and (3) in the peak season of transportation, the capacity cannot meet the demand. Through comparison with indicators in the literature, it was found that "level of specialization of loading and unloading equipment", "proportion of piggyback transport", "time of order response", "ratio of on-time delivery", "efficiency of loading and unloading

operations", "degree of information sharing", and "automatic design and generation system for the whole logistics scheme" do not appear in the literature index statistics. Therefore, these indicators were added to the final indicator system.

**Table 2.** Topics and outline of the interview.

| Interview Topics | Interview Outlines |
|---|---|
| Current status of this company's container multimodal transport | What is the number of customers in recent years? What kind of business are customers? Is your relationship with customers stable? What are the types of goods that have recently adopted container multimodal transport? What are the differences between container multimodal transport and bulk cargo transportation requirements (costs, timeliness, business handling, etc.)? What are customers most concerned about? What are some of the issues that you communicate more with your customers? What are the requirements of customers that you can meet well, and which are not yet able to meet customer needs? Do you give customers an open service commitment? |
| Coordination of "software and hardware" for container multimodal transport | Are the existing facilities and equipment complete? Has the technical equipment reached the standardization level? Does the technical equipment match the use of the goods? Are the staff efficient? What are the existing port loading and unloading methods? How long does it take for goods to be stored at the station? During the operation, what are the key links that affect the connection of intermodal transportation? |
| Analysis of the competitiveness of this company's container multimodal transport | What are the main ways or combinations of competition with your business? What service links do you think need to be improved in the current multimodal transport production process? What do you think is the most urgent task to improve the competitiveness of the entire supply chain of multimodal transport? |

*4.3. Selection of Indicators Based on Policy Documents, Regulations, and Plans*

Comprehensive transportation, as one of the main sources of carbon emissions, is receiving increasing attention from government departments. A series of China's policy documents, regulations, and plans also focus on adjusting the transport structure and promoting the coordination of various modes of transport. In order to ensure that the indicators are in line with the economic and political development trends and are time-effective and statutory, we selected 36 policy documents and industry reports related to container and multimodal transport. Considering the unstructured nature of the text, we used content analysis to reveal the information in the file content that was relevant to the study of this paper. By selecting an analysis unit, formulating an analysis system, and conducting quantitative analysis, we obtained 236 effective analysis categories. Through screening and induction, 10 indicators were formed, as shown in Table 3. We found that these indicators appear in literature indicators and in-depth interviews, indicating that they are in line with economic and political development trends.

**Table 3.** Content analysis results of policy documents, regulations, and plans.

| Number | Indicators | Frequency | Representative Policy Documents | Example |
|--------|-----------|-----------|-------------------------------|---------|
| 1 | Information resource sharing | 29 | "13th Five-Year Plan" for Comprehensive Transportation Services | Guide multimodal transport affiliated enterprises to strengthen information system interconnection and collaboration and promote multimodal transport information resource sharing. |
| 2 | Transportation organization convergence | 23 | Development Planning for Modern Comprehensive Transport Traffic System of "The 13[th] Five-Year Plan" | Realizing the effective connection of various transportation mode standards can improve the comprehensive transportation service support capability and level. |
| 3 | Energy saving and emission reduction level | 17 | Notice of the General Office of the State Council on Promoting the Adjustment of Transport Structure | Increase support for the construction of demonstration project logistics parks (freight hubs), the promotion and application of new energy vehicles, and the construction of green logistics smart service platforms. |
| 4 | Intelligent technical equipment | 15 | Container Multimodal Transport Development Report 2018 | Multimodal transport loading units, loading equipment and carrying equipment are forming system solutions through intelligent technology innovation. |
| 5 | Unification of electronic documents | 14 | National Logistics Hub Layout and Construction Plan | Research the implementation of electronic unified documents for container multimodal transport among national logistics hubs and strengthen the exchange and sharing of document information. |
| 6 | Complete level of supporting facilities | 12 | Guiding Opinions of the Central Committee of the Communist Party of China on Carrying out Quality Improvement Actions | Consolidate the national quality infrastructure and accelerate the construction of the national standard system. |
| 7 | Clearance efficiency | 12 | Logistics industry cost reduction and efficiency improvement special action plan 2016–2018" | Implement information exchange, mutual recognition of supervision, mutual assistance in law enforcement, advance the "single window" construction and "one-stop operation" reforms, and improve customs clearance efficiency. |
| 8 | Loading and unloading efficiency | 10 | National Logistics Hub Layout and Construction Plan | Improve the efficiency of cargo reloading between different modes of transport, improve operational efficiency and integrated organizational level. |
| 9 | Use of piggyback transport | 4 | Container Multimodal Transport Development Report 2018 | The function of rotating or panning the back transport vehicle can load and unload the whole train without removing the hook, and the operation efficiency is high. |
| 10 | Freight capacity | 4 | "The 13[th] Five-Year Plan" for Railway | The freight transportation capacity basically meets the transportation needs of energy and resources across the region. |

*4.4. Establishing Indicator System*

Due to the small amount of Chinese and English literature that can be analyzed for content, establishing the indicator system of this study focused more on the indicators obtained from qualitative research and analysis of policy documents. According to the structure of the container multimodal transport system, three major categories of indicators were screened, and were effectively combined and redefined to determine the preliminary indicator system based on the foregoing analysis, as shown in Table 4.

**Table 4.** Evaluation indicators system for the synergy degree of the container multimodal transport system.

| Subsystem | Order Parameters (Indicators) | Sequence | Property | Calculation or Definition |
|---|---|---|---|---|
| Facilities and equipment subsystem | Dedicated railway line at the port | 1 | + | With dedicated line: 1; Without dedicated line: 0 |
| | Matching degree of capacity and demand | 2 | + | The percentage of demand that can be met during peak season |
| | Level of specialization of loading and unloading equipment | 3 | + | The percentage of dedicated loading and unloading equipment |
| | Intelligence degree of technical equipment | 4 | + | The percentage of technical equipment with computer independent data control capability |
| | Proportion of piggyback transport | 5 | + | The percentage of containers transported by piggyback |
| Organization and management subsystem | Standardization of multimodal organization processes | 6 | + | With structured processes across enterprises: 1; companies rely entirely on manual connections: 0 |
| | Time of order response | 7 | − | Time from the customer's request to the carrier's reply "successful acceptance" |
| | Order processing time | 8 | − | Time from the carrier's response to the "successful acceptance" to the shipment |
| | Transfer time | 9 | − | Transportation mode change time (Rail–sea transportation) |
| | Efficiency of loading and unloading operations | 10 | − | Total labor hours for loading and unloading a standard container |
| | Ratio of on-time delivery | 11 | + | The percentage of orders arriving on time (given unit time) |
| | Limitation of vehicle emission levels entering the site | 12 | + | Restricted requirements: 1; No restrictive requirements: 0 |
| Business operation subsystem | Organization of collaborative platform for short-distance transportation companies | 13 | + | With platform organization: 1; Without platform organization: 0 |
| | The percentage of the usage of the single contract | 14 | + | The percentage of the usage of the single contract |
| | Customs processing time | 15 | − | Time spent in customs clearance |
| | Cargo damage claim time | 16 | − | Time from damage to payment of claims |
| | Automatic design and generation system for the whole logistics scheme | 17 | + | With automatic generation system: 1; No/Unused automatic generation system: 0 |
| Information interaction subsystem | Information tracking ability | 18 | + | The percentage of orders tracking goods throughout the process |
| | Degree of information sharing | 19 | + | The percentage of the total number of information sharers |
| | Multimodal Public Information Platform | 20 | + | With public information platform: 1; Without public information platform: 0 |

**Note:** "+" represents positive sequence parameter; "−" represents negative sequence parameter.

## 5. Applying Evaluation Model for Synergy Degree of Container Multimodal Transport System

Based on the evaluation indicators for the synergy degree of container multimodal transport, Section 5.1 establishes a synergy degree evaluation model, and Section 5.2 shows the data processing, including data standardization and weight determination. Finally, the application and analysis of China G port is taken as an example in Section 5.3.

### 5.1. Synergy Degree Evaluation Model of Container Multimodal Transport System

Container multimodal system S is a composite system composed of four subsystems, $S = \{S_1, S_2, S_3, S_4\}$, where $S_i$ is the $i$th subsystem, $i = 1, 2, 3, 4$. $S_1, S_2, S_3, S_4$ are business operations, organization and management, facilities and equipment, and information interaction subsystems, respectively. The indicator in each subsystem is defined as the cooperative order parameter $a_{ij}$. Suppose that a certain moment is $T_n (n = 1, 2, \ldots, m)$, and $T_1$ is the initial time. The degree of order $U_{T_n}$ of $a_{ij}$ has the following formula [43]:

$$U_{T_n}(a_{ij}) = \begin{cases} \frac{a_{ij}^{T_n} - \alpha}{\beta - \alpha}, & \left(\text{where } a_{ij} \text{ is a positive order parameter}\right) \\ \frac{\beta - a_{ij}^{T_n}}{\beta - \alpha}, & \left(\text{where } a_{ij} \text{ is a negative order parameter}\right) \end{cases} \tag{1}$$

where $a_{ij}^{T_n}$ refers to the specific value of $a_{ij}$ at $T_n$, $\alpha$ and $\beta$ are the minimum and maximum value of $a_{ij}$ during the entire reporting period, respectively. The numerical value of $U_{T_n}(a_{ij})$ can explain the contribution of $a_{ij}$ to $S_i$'s degree of order, and its value is proportional to the degree of order. We determined the contribution value of all index parameters to synergy degree $S_i$ of the subsystem, and integrated by linear weighted summation:

$$U_{T_n}(S_i) = \sum_{j=1}^{k} \partial_{ij} U_{T_n}(a_{ij}), \ \partial_j > 0, \ \sum_{j=1}^{k} \partial_j = 1 \tag{2}$$

where $\partial_{ij}$ refers to the weight of the $j$th indicator parameter in subsystem $S_i$. The larger the value of $U_{T_n}(S_i)$, the higher the degree of synergy at time point $T_n$. The total degree of synergy from $T_1$ to $T_m$ is calculated as follows:

$$U_{T_n} = \sqrt[4]{\omega \sum_{i=1}^{4} \left| U_{T_n}(S_i) - U_{T_1}(S_i) \right|} \tag{3}$$

$$\omega = \frac{\left( U_{T_n}(S_i) - U_{T_1}(S_i) \right)_{min}}{\left| U_{T_n}(S_i) - U_{T_1}(S_i) \right|_{min}} \tag{4}$$

In the formula, $U_{T_n}$ is the degree of coordination at $T_n$. When it is positive, it indicates that each indicator parameter in the container multimodal transport system achieves coordinated development through mutual influence and coordination. The larger the value of $U_{T_n}$, the higher the degree of coordination of the system and the better the coordination. When $U_{T_n}$ is negative, it means that at least one subsystem is developing in a disorderly direction.

### 5.2. Normalization of Data

First, evaluating the degree of synergy must involve measuring and calculating data of different subsystems and indicator parameters, so the standard deviation formula is used to standardize the objective data of each coordinate index parameter. $a_{ij}^{T_n}$ refers to the specific value of $a_{ij}$ at $a_{ij}$, $\bar{a}_{ij}$ is

the average value of each point in the measurement period ($T_1$–$T_n$), and $R_i$ is the sample standard deviation of subsystem $S_i$. The data normalization formula is:

$$\dot{a}_{ij}^{T_n} = \frac{a_{ij}^{T_n} - \bar{a}_{ij}^{T_n}}{R_{ij}} \tag{5}$$

$$R_{ij} = \sqrt{\frac{1}{m-1}\sum_{j=1}^{m}\left(a_{ij}^{T_n} - \bar{a}_{ij}\right)^2} \tag{6}$$

Secondly, the weight of the indicators is also involved in the specific measurement. The weight reflects the influence of different indicator parameters on the synergy of the system. The larger the weight, the greater the degree of effect of the indicator parameter on the synergy degree of the system. It is unavoidable that the foregoing article included certain supervisory factors in the process of establishing the index, so we chose the objective weighting method–entropy weighting method when empowering. We used the entropy weight method to determine the weight of each index parameter, substituted the standardized data and specific weights into the evaluation model, and calculated the degree of synergy. The basic idea of the entropy weight method is to determine the objective weight according to the size of the index variability. In general, if the information entropy of an indicator is smaller, it indicates that the indicator deserves more variation and the more information it provides, the greater the role it can play in comprehensive evaluation, and the greater its weight. On the contrary, the larger the information entropy of an indicator, the smaller the degree of variation of the indicator value, the smaller the amount of information provided, the smaller the role it plays in the comprehensive evaluation, and the smaller its weight.

First, the data of each indicator is normalized to obtain $x\prime_{ij}$, where there are i time units and j indicators. Then we calculate the index value proportion $P_{ij}$ of the item i in the index j to obtain the entropy value e of the j-th index, where $k = \frac{1}{\ln(n)} > 0$, which satisfies $e_j \geq 0$. Finally, the redundancy $d_j$ of the information entropy is calculated to obtain the weight value $w_j$ of each index.

$$x'_{ij} = \begin{cases} \dfrac{x_{ij}-\min\{x_{ij},...,x_{nj}\}}{\max\{x_{1j},...,x_{nj}\}-\min\{x_{1j},...,x_{nj}\}}, & \left(\text{where } x_{ij} \text{ is a positive order parameter}\right) \\ \dfrac{\max\{x_{1j},...,x_{nj}\}-x_{ij}}{\max\{x_{1j},...,x_{nj}\}-\min\{x_{1j},...,x_{nj}\}}, & \left(\text{where } x_{ij} \text{ is a negative order parameter}\right) \end{cases} \tag{7}$$

$$P_{ij} = \frac{x\prime_{ij}}{\sum_{i=1}^{n} x\prime_{ij}} \tag{8}$$

$$e_j = -k\sum_{i=1}^{n} P_{ij}\ln(P_{ij}), \left(P_{ij} = 0, e_j = 0\right) \tag{9}$$

$$w_j = \frac{d_j}{\sum_{j=1}^{m} d_j}, d_j = 1 - e_j \tag{10}$$

*5.3. Application*

In order to test the feasibility and rationality of the synergy degree evaluation system and model of the container multimodal transport system, we took China's G port as an example and used its original data for analysis. The container multimodal transport chain includes freight forwarders, railway transport companies, shipping companies, ports, cargo owners (customers), and other business entities, all of which can be used as evaluation objects. China's G port has great potential for development and can be observed for the coordinated development of the entire intermodal transport chain.

G port is in a developed coastal area, and its seaport includes four major port areas. In 2018, the container throughput of G port exceeded 20 million TEUs, and the container transportation development momentum was strong. Road and inland waterway transportation remain the main mode of container collection and distribution at the port. The proportion of sea–rail transport is relatively small, because the new port area can reach the special railway line in the old port area, which is connected to the Beijing–Guangzhou and Guangzhou–Shenzhen line, only by boat to achieve sea–rail combined transport. The second special railway line goes directly to Xingang District and will not be put into use until 2020.

In the data acquisition and calculation, we fully considered that the G port container multimodal transport has strong potential and rapid development. If half a year or one year is used as the time unit, the results of data acquisition and analysis may be very rough; if one month is used as the time unit, the amount of data acquisition and calculation is too large, and the analysis results are scattered. Therefore, we chose quarterly as the time unit. Based on the elements of the container multimodal transport system and the established indicator system, we obtained the raw data for the 16 quarters of G port in 2015–2018. According to data normalization and ordered calculation formulas, all the data obtained were converted into the data presented in Tables A2 and A3. Due to certain subjective factors in the process of establishing indicators, the objective weighting–entropy weighting method was chosen to determine the weight. Through calculation, the weights of 20 indicators under the four subsystems were obtained, as shown in Table 5.

**Table 5.** Weights of indicators.

| Order Parameter | 1 | 2 | 3 | 4 | 5 | 6 | 7 | 8 | 9 | 10 |
|---|---|---|---|---|---|---|---|---|---|---|
| Weight | 0.106 | 0.135 | 0.200 | 0.199 | 0.361 | 0.313 | 0.099 | 0.099 | 0.084 | 0.120 |
| Order Parameter | 11 | 12 | 13 | 14 | 15 | 16 | 17 | 18 | 19 | 20 |
| Weight | 0.169 | 0.116 | 0.100 | 0.289 | 0.085 | 0.073 | 0.453 | 0.272 | 0.123 | 0.606 |

The standardized data and index weights were integrated into the container multimodal transport synergy measurement model, and the contribution values and total synergy of all index parameters to the synergy of subsystem *Si* were calculated, as shown in Table 6. The development trend of synergy is shown in Figure 4.

**Table 6.** Synergy degree.

| Time | Facilities and Equipment Subsystem | Organization and Management Subsystem | Business Operation Subsystem | Information Interaction Subsystem | Total Synergy Degree |
|---|---|---|---|---|---|
| $T_1$ | 0.0451 | 0.0000 | 0.0000 | 0.0000 | - |
| $T_2$ | 0.1510 | 0.1500 | 0.0734 | 0.0615 | 0.7305 |
| $T_3$ | 0.1285 | 0.1831 | 0.0734 | 0.0615 | 0.7959 |
| $T_4$ | 0.1059 | 0.0662 | 0.2585 | 0.0615 | 0.8177 |
| $T_5$ | 0.3932 | 0.4455 | 0.3740 | 0.0615 | 1.0529 |
| $T_6$ | 0.3932 | 0.4455 | 0.2585 | 0.0615 | 1.0273 |
| $T_7$ | 0.3932 | 0.7484 | 0.2585 | 0.1293 | 1.1038 |
| $T_8$ | 0.3481 | 0.5016 | 0.2585 | 0.2312 | 1.0666 |
| $T_9$ | 0.3747 | 0.9270 | 0.2585 | 0.3637 | 1.1708 |
| $T_{10}$ | 0.6451 | 0.9270 | 0.2585 | 0.3637 | 1.2108 |
| $T_{11}$ | 0.6451 | 0.6140 | 0.2585 | 0.3637 | 1.1641 |
| $T_{12}$ | 0.6451 | 0.5578 | 0.3740 | 0.3637 | 1.1734 |
| $T_{13}$ | 1.0000 | 0.9669 | 0.5472 | 0.9693 | 1.3617 |
| $T_{14}$ | 1.0000 | 0.6539 | 1.0000 | 0.9693 | 1.3753 |
| $T_{15}$ | 1.0000 | 0.6539 | 1.0000 | 1.0000 | 1.3783 |
| $T_{16}$ | 1.0000 | 1.0000 | 0.7113 | 1.0000 | 1.3837 |

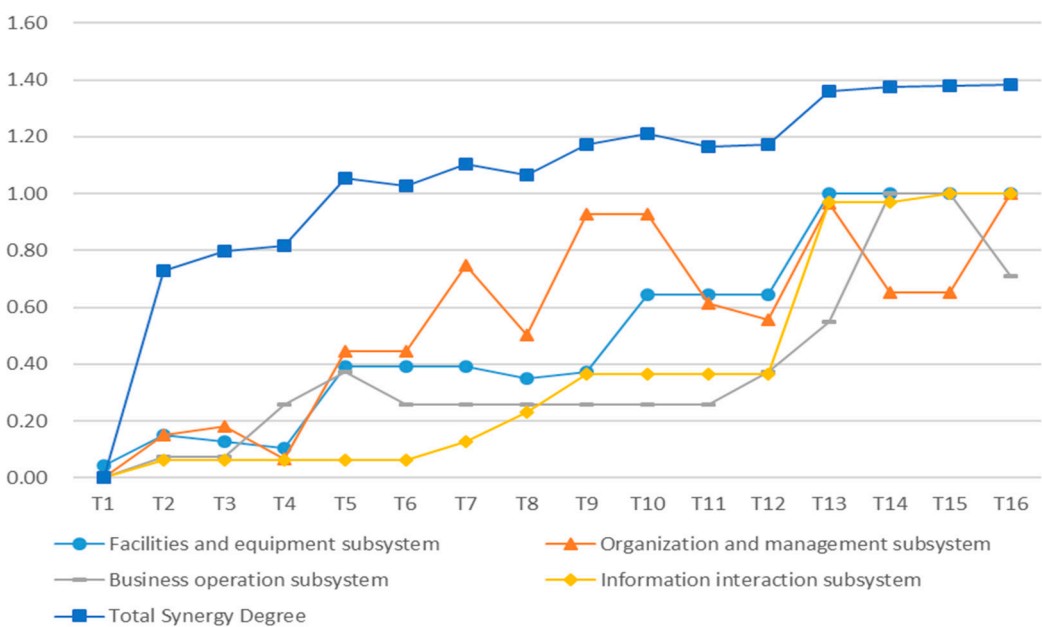

**Figure 4.** Development trend of synergy degree.

(1) Analysis of synergy degree of facilities and equipment subsystem

As shown in Figure 2, during the observation period, the synergy degree of the facilities and equipment subsystem was positive and increased stepwise. In the time periods T1–T2, T4–T5, and T12–T13, there were declines in different degrees. Other phases slightly increased and dropped. During the observation period, the dedicated railways in H port (one of the four major ports) began to be used on a large scale, and the proportion of piggyback transport and the specialization level of loading and unloading equipment also gradually increased.

Overall, the facilities and equipment subsystem was optimized and improved. However, the matching of transportation capacity and demand and the fluctuation of the level of technical equipment intelligence still affected the steady increase of system collaboration.

(2) Analysis of synergy degree of organization and management subsystem

As shown in Figure 2, the synergy degree of the organization and management subsystem was positive, showing a convoluted rise and a sharp rise and fall. In the time periods T3–T4, T7–T8, T10–T12, and T13–T14, there were declines in different degrees: (1) in T3–T4, order processing time and transfer time slightly increased; (2) in T7, the multimodal transportation process was standardized, while in T8 manual connection was used to organize the process, and T13–T14 was the same; (3) T10–T12 also had fluctuations in standardization, and the punctuality rate decreased slightly at T12. In the three annual nodes of T4–T5, T6–T7, T8–T9, T12–T13, and T15–T16, the normal trend resumed, and the synergy degree of the organizational management subsystem was higher than that of the other three subsystems during most of the observation period. Overall, growth coincided with volatility.

The overall capacity of the multimodal organization of the port was improved, but the fluctuations in its standardization revealed that the standardization progress in the entire process of intermodal transportation was not smooth. The standardization of the process plays an important role in the organization and management system. Managers should actively participate in and improve the standardization services of the process, encourage multimodal operators to participate in the hub operation, make the hub more closely join the multimodal transport organization chain, and gradually replace the standardized process with manual connection to achieve deep integration and smooth transfer of processes.

(3) Analysis of synergy degree of business operation subsystem

As shown in Figure 2, the synergy degree of the business operation system was positive during the observation period, but there were several fluctuations. The percentage of use of single contracts declined in the time periods T5–T6 and T15–T16, resulting in a simultaneous decrease of synergy degree. Indicator data did not change significantly in the time period T6–T11, so the synergy degree did not fluctuate significantly. Although the overall growth is obvious due to the decline in the percentage of single contracts in the last stage, the synergy degree dropped sharply.

The coordination ability of business operations of the G port container multimodal transport improved during the observation period, but the development trend was not ideal. In the Implementation Plan for Creating a Good Market Environment to Promote the Integration and Development of Transportation and Logistics issued by the National Development and Reform Commission of China in 2016, implementation of single contracts in the entire process was mentioned, but the percentage of use of single contracts at G port fluctuates, indicating that its development is unstable and has not been popularized throughout the entire multimodal transport process. It also revealed the existence of joint liability, cost settlement, and inconsistent delivery conditions among the carriers in the entire process of multimodal transportation. The use of single contracts is the most difficult point in the construction of the entire system. It cannot be achieved by a single carrier. Each multimodal host should focus on solving the problem of inconsistent documents in domestic trade transportation and facilitate the conversion to multiple transport modes. China should cooperate with neighboring countries to promote the use of inland international multimodal transport bills of lading to reduce the risk of multimodal transport through the materialization of documents.

(4) Analysis of synergy degree of information interaction subsystem

As shown in Figure 2, during the observation period, the synergy degree of the information interaction subsystem was positive and showed an upward trend. The observation period was divided into three parts: T1–T6, in which the step growth of T1–T2 was relatively slow and T2 showed no significant growth; T6–T12, in which T6–T9 grew slowly and T9 began to show no significant growth; and T12–T16, in which T12–T13 (the last quarter of 2017) increased significantly and growth slowed from T13.

In general, the growth is obvious. Compared with the synergy degree of the organization management subsystem, the information interaction subsystem significantly improved every year. Compared with the step-up of the facility equipment subsystem, the step-up of the information interaction subsystem is more stable and has not seen a drop. During the observation period, the coordination of the information interaction was ideal, and the ability improved. The multimodal public information platform was realized from scratch, the ability to track information has improved significantly, and information is gradually shared among participants. G port container multimodal transport already has an open collaborative information exchange system.

(5) Analysis of total synergy degree

As shown in Figure 2, the overall synergy degree was positive. It can be divided into two phases: in T1–T2, the overall synergy degree of the enterprise experienced explosive growth, and in T2–T16, it remained stable. Although there were fluctuations, the development was relatively stable and slow. The efficiency and quality of enterprises improved in this period, but the growth rate was small. From the development trend of total synergy degree, it can be concluded that during the observation period, G port synergy gradually developed and stabilized from an unstable state, but the overall development was slow and the growth rate was small.

Through analysis, we find that the overall synergy degree in the container sea–rail multimodal operation of G port is relatively slow and inefficient. Taking 2017–2018 as an example, in 2017, the container throughput exceeded 20 million TEUs, but the container transportation volume completed by combined sea–rail transportation was only 50,000 TEUs, and from January to November

2018, the container throughput was 20 million TEUs and the combined sea–rail transportation was only 56,000 TEUs. The various parties are not closely coordinated in organizational management, mature multimodal organization processes have not been formed, and manual connection is still being used. The business operation is not smooth, and the use of single contracts has not been realized. These are all important issues that affect the efficiency of combined sea–rail transport at G port.

The previous analysis of the G port problem and the study of the general development status of multimodal transport not only provide a certain reference for business managers, but also bring benefits to policy makers. For logistics service providers, the results of the degree of synergy measurement can enable them to know which aspects they can work on to improve the multimodal transport synergy more effectively, enhance their competitiveness, and contribute to the reduction of carbon emissions in social logistics. For freight customers, they can evaluate the multimodal transport supply chain based on this indicator system and choose effective, low-carbon suppliers. For policy makers, they can understand the blocking points and uncertainties in actual operation through specific research on multimodal transport projects. This can help them formulate effective multimodal transport policies while ensuring economic development. This will help guide logistics companies and cargo owners to achieve smoother multimodal transport.

To promote the development of multimodal transport, the transportation department can formulate relevant policies from three aspects: standardization, normalization, and infrastructure construction. First, the transportation department should strengthen the standardization of multimodal transport development, which includes standardization of infrastructure, equipment, tools, conversion devices, data and documents. Second, normalization of operations at freight yards and the market behavior are useful. Third, the transportation department should carry out infrastructure construction, including the construction of public information platforms and multimodal transport stations. We consider that multimodal transport can effectively reduce carbon emissions. Other government departments can control the use of carbon-intensive transportation through taxation policies. They can also promote awareness of low-carbon behaviors among citizens through publicity and education.

From the perspective of the development trend of international containers in the world, vigorously promoting multimodal transport, especially sea–rail transport, is the only way for China's container transportation to develop rapidly, and it is also an inevitable choice for the development of low-carbon logistics. A government can take the development of sea–rail intermodal transport as a breakthrough, work hard to promote the construction of an integrated container multimodal transport network, and move from a single road highway to a sea–rail intermodal transport, a public-rail intermodal transport, and establish a multi-channel, multi-directional intermodal transport network.

## 6. Conclusions

In this paper, we combined the concept of synergy with the field of integrated transportation. We paid attention to the bottleneck in the development of China's container multimodal transport, the problem of unsmooth coordination, and established a container multimodal transport system evaluation system and model of synergy.

We extended the existing research on container multimodal transport from two dimensions of research content and method. First, we found that researchers and managers have recognized the importance of container intermodal transport operations, but current studies do not evaluate their synergy. We filled this gap by applying synergy theory to the research of container multimodal transport, analyzing and evaluating it from the perspective of the whole process. Secondly, in the actual evaluation process, we did not use questionnaires to determine the true development of China's container multimodal transport, because that could have resulted in receiving many invalid questionnaires. Instead, we selected representative companies and customers and innovatively used in-depth interviews and grounded theory, commonly used in social sciences, to construct an index system and model to evaluate the degree of collaboration in the container multimodal transport system. On this basis, we conducted a synergy evaluation based on the development of China's G

port sea–rail transport from 2015 to 2018, and found that the intermodal operation of G port, which is one of the top 10 ports in China, maintained synergy, but the growth was slow. This can serve as a warning to managers to keep abreast of national policies and economic development trends, and use intelligent technology and reasonable operating models to improve the operation level of the container multimodal transport system.

The coordination degree evaluation model was used to measure the synergy degree of container multimodal transport during the observation period, find the links that need improvement, and formulate countermeasures. At the same time, we noticed that it was impossible to predict the level of coordination outside the observation period and determine how to stabilize the synergy degree. Therefore, this is our next research direction.

**Author Contributions:** Conceptualization, writing—review and editing, X.F.; methodology and writing—original draft preparation, Z.J.; supervision, Z.C. and W.C.; investigation, C.C.; data curation, J.G.; All authors have read and agreed to the published version of the manuscript.

**Funding:** This research was financially supported by National Key R & D Program of China (No. 2017YFB1201300) and Science Progress and Innovation Program of Hunan DOT, 201949.

**Conflicts of Interest:** The authors declare no conflict of interest.

## Appendix A

Table A1. Selection of indicators based on in-depth interviews with container multimodal transport service providers and customers.

| Category | Source Data Statements (two Samples Taken as Examples) |
|---|---|
| Intelligence degree of technical equipment | A7: We have been paying attention to the intelligent problem of the equipment, and the embedded algorithms in the equipment realize independent maintenance. A10: Transportation equipment has not achieved widespread intelligence. The intelligence of the most basic sports units and freight models will directly improve operating efficiency. |
| Matching degree of railway lines of the port | A2: The railway line for bulk cargo is still being planned. At that time, when the railway was being repaired, it was not repaired there. In actual operation, the railway and transportation conditions can be matched to operate efficiently. A7: The hot metal transportation depends on whether there is a professional line, and now it is necessary to vigorously promote the professional line into the port area. This is still a shortcoming. |
| Matching degree of capacity and demand | A1: From Guangzhou to Urumqi and Shenyang, the cost is more cost-effective than the existing model. We consider whether to have such capability every time we serve. A3: Railways are related to economic development. It is proposed that the development of southwestern construction will consume a lot of money, but the capacity of Chengdu Bureau itself is not very large. |
| Time of order response | A4: The ability to respond quickly to "successful acceptance" has a great impact on the organization plan of intermodal transport. A6: From the perspective of the shipper, I hope that, if possible, the customer's demand will be accepted immediately, and the organization and coordination can be operated quickly. |
| Limitation of vehicle emission levels entering the site | A6: On the issue of environmental protection, there are now requirements for the emissions of cars entering freight stations. A11: Isn't it the conversion of old and new kinetic energy, that is, the vehicles of National III and National IV have to be eliminated, the emissions are not up to standard, and gasoline vehicles of National V are used. |
| Automatic design generation system for the whole logistics solution | A6: I have to say that from 16 years on, we began to use the full logistics solution to automatically design and generate the system, and the overall efficiency has been greatly improved. A9: Not only the middle link, but also the end-to-end needs to be quickly linked in order to coordinate the parties. If the scheme design can be generated automatically, the degree of coordination will increase. |

<div align="center">

**Table A1.** *Cont.*

</div>

| Category | Source Data Statements (two Samples Taken as Examples) |
|---|---|
| Standardization of multimodal organization processes | A1: The organization process of intermodal transportation is sometimes perfect or imperfect, which affects the connection with other links.<br>A11: There are countless troubles caused by organizational processes. Some businesses have well-structured processes, so the process is smooth. However, some businesses rely solely on human integration and are inefficient. |
| Organization of collaborative platform for short-distance transportation companies | A4: There is about five kilometers between the first and third phases of our project, so there is always a short-distance transportation, which involves costs.<br>A7: If there is a complete collaborative platform organization for short-haul transportation companies, it will bring many benefits. |
| Order processing time | A1: After replying "successfully accepted" to the customer, we will process this order as soon as possible, including the case adjustment and viewing plan.<br>A6: If this process is slow, it will directly affect the following collaboration processes. |
| Degree of information sharing | A1: A lot of information data is not exchanged, the amount of information data exchange is small, and the coverage of information data is narrow.<br>A5: Some data and information may be available to my company, but not necessarily your company. We also don't want to give you information, because we will worry about network security issues. |
| Transfer time | A2: In order to improve the overall level of multimodal transport of containers, we need to transfer from the container, including the handling of straddle carriers.<br>A5: There are many problems in the transfer of transportation. The inconsistency between the two parties will reduce the efficiency. |
| Use of single contracts | A7: As a container carrier of hot metal, we hope that the other party will standardize electronic documents, but it is difficult.<br>A11: The container multimodal transport unified document is missing. |
| Cargo damage claim time | A7: Customer maintenance is very important, so consider the cooperation between the carrier and the customer. The customer has a high demand for compensation for cargo damage.<br>A8: Once the goods are damaged, whether the procedures can be compensated to the customer as soon as possible will affect the next operation. |

**Table A1.** *Cont*.

| Category | Source Data Statements (two Samples Taken as Examples) |
|---|---|
| Customs processing time | A3: The administrative efficiency of customs clearance really affects the efficiency. A14: Shortening the time in declaration, inspection, and release not only saves customs clearance costs and improves efficiency, but also provides support for improving the timeliness of trains. |
| Efficiency of loading and unloading operations | A3: The terminal visited today is the loading and unloading module. The loading speed and efficiency are very important. A4: The rail gantry crane is directly loaded and unloaded and put into the yard, so the loading and unloading is fast and efficient. |
| Ratio of on-time delivery | A4: There is a time window for delivery. It can't be delivered early or late. Delivery on time is very important. A8: The whole process means that every link is done well. Many people think that the goods are delivered. But if the delivery is not on time, then the previous link is done in vain. |
| Proportion of piggyback transport | A5: To be honest, the piggyback method can save time and make the whole connection process smooth. A6: Piggyback is a way of grouping or unitizing, and the degree of coordination is high. |
| Level of specialization of loading and unloading equipment | A3: In the port, the replacement of cargo, the layout of the rail gantry crane, the layout of the yard, and the special equipment for collecting trucks are all the key points to achieve coordination and cooperation. A4: To realize the rapid deployment of yards and stations, it is required that the special supporting facilities of cranes, trucks, and yards are reasonable. |
| Information tracking ability | A4: Many customers hope to achieve full tracking of the goods, but we can't. A5: If we can meet the tracking problem of information, we can grasp the cargo dynamics in real time, and the speed of cargo flow and connection can be accelerated. |
| Multimodal public information platform | A10: Build a shared platform for multimodal transport and develop a multimodal transport operation management system. A11: There is another important issue: the container multimodal transport information management platform is almost blank. |

**Table A2.** Normalization of data.

| | T1 | T2 | T3 | T4 | T5 | T6 | T7 | T8 | T9 | T10 | T11 | T12 | T13 | T14 | T15 | T16 |
|---|---|---|---|---|---|---|---|---|---|---|---|---|---|---|---|---|
| $\dot{a}_{11}^{T_n}$ | −3.7422 | 0.2495 | 0.2495 | 0.2495 | 0.2495 | 0.2495 | 0.2495 | 0.2495 | 0.2495 | 0.2495 | 0.2495 | 0.2495 | 0.2495 | 0.2495 | 0.2495 | 0.2495 |
| $\dot{a}_{12}^{T_n}$ | −0.7711 | −0.7711 | −1.3075 | −1.8439 | 0.3017 | 0.3017 | 0.3017 | −0.7711 | −0.2347 | −0.2347 | −0.2347 | −0.2347 | 1.3746 | 1.3746 | 1.3746 | 1.3746 |
| $\dot{a}_{13}^{T_n}$ | −1.3236 | −1.3236 | −1.3236 | −1.3236 | −0.3309 | −0.3309 | −0.3309 | −0.3309 | 0.3309 | 0.3309 | 0.3309 | 0.3309 | 1.3236 | 1.3236 | 1.3236 | 1.3236 |
| $\dot{a}_{14}^{T_n}$ | −1.3300 | −1.3300 | −1.3300 | −1.3300 | 0.3069 | 0.3069 | 0.3069 | 0.3069 | −0.3069 | −0.3069 | −0.3069 | −0.3069 | 1.3300 | 1.3300 | 1.3300 | 1.3300 |
| $\dot{a}_{15}^{T_n}$ | −0.8235 | −0.8235 | −0.8235 | −0.8235 | −0.8235 | −0.8235 | −0.8235 | −0.8235 | −0.8235 | 0.7576 | 0.7576 | 0.7576 | 1.2846 | 1.2846 | 1.2846 | 1.2846 |
| $\dot{a}_{21}^{T_n}$ | −0.6437 | −0.6437 | −0.6437 | −0.6437 | −0.6437 | −0.6437 | 1.4162 | −0.6437 | 1.4162 | 1.4162 | −0.6437 | −0.6437 | 1.4162 | −0.6437 | −0.6437 | 1.4162 |
| $\dot{a}_{22}^{T_n}$ | 1.9243 | −0.3564 | 1.9243 | −0.3564 | −0.3564 | −0.3564 | 1.9243 | −0.3564 | −0.3564 | −0.3564 | −0.3564 | −0.3564 | −0.3564 | −0.3564 | −0.3564 | −1.4967 |
| $\dot{a}_{23}^{T_n}$ | 2.0012 | 2.0012 | −0.4618 | 2.0012 | −0.4618 | −0.4618 | −0.4618 | −0.4618 | −0.4618 | −0.4618 | −0.4618 | −0.4618 | −0.4618 | −0.4618 | −0.4618 | −0.4618 |
| $\dot{a}_{24}^{T_n}$ | 2.5504 | −0.3643 | −0.3643 | 2.5504 | −0.3643 | −0.3643 | −0.3643 | −0.3643 | −0.3643 | −0.3643 | −0.3643 | −0.3643 | −0.3643 | −0.3643 | −0.3643 | −0.3643 |
| $\dot{a}_{25}^{T_n}$ | 1.5524 | 1.5524 | 1.5524 | 1.5524 | −0.2218 | −0.2218 | −0.2218 | −0.2218 | −0.2218 | −0.2218 | −0.2218 | −0.2218 | −1.1088 | −1.1088 | −1.1088 | −1.1088 |
| $\dot{a}_{26}^{T_n}$ | −1.1025 | −1.1025 | −1.1025 | −1.1025 | −1.1025 | −1.1025 | −0.3969 | −0.3969 | 1.0143 | 1.0143 | 1.0143 | 0.3087 | 1.0143 | 1.0143 | 1.0143 | 1.0143 |
| $\dot{a}_{27}^{T_n}$ | −1.6599 | −1.6599 | −1.6599 | −1.6599 | 0.5533 | 0.5533 | 0.5533 | 0.5533 | 0.5533 | 0.5533 | 0.5533 | 0.5533 | 0.5533 | 0.5533 | 0.5533 | 0.5533 |
| $\dot{a}_{31}^{T_n}$ | −2.0012 | −2.0012 | −2.0012 | 0.4618 | 0.4618 | 0.4618 | 0.4618 | 0.4618 | 0.4618 | 0.4618 | 0.4618 | 0.4618 | 0.4618 | 0.4618 | 0.4618 | 0.4618 |
| $\dot{a}_{32}^{T_n}$ | −0.5884 | −0.5884 | −0.5884 | −0.5884 | 0.4026 | −0.5884 | −0.5884 | −0.5884 | −0.5884 | −0.5884 | −0.5884 | 0.4026 | 1.8890 | 1.8890 | 1.8890 | −0.5884 |
| $\dot{a}_{33}^{T_n}$ | 2.5504 | −0.3643 | 2.5504 | −0.3643 | −0.3643 | −0.3643 | −0.3643 | −0.3643 | −0.3643 | −0.3643 | −0.3643 | −0.3643 | −0.3643 | −0.3643 | −0.3643 | −0.3643 |
| $\dot{a}_{34}^{T_n}$ | 3.1688 | 1.7604 | −0.3521 | −0.3521 | −0.3521 | −0.3521 | −0.3521 | −0.3521 | −0.3521 | −0.3521 | −0.3521 | −0.3521 | −0.3521 | −0.3521 | −0.3521 | −0.3521 |
| $\dot{a}_{35}^{T_n}$ | −0.4618 | −0.4618 | −0.4618 | −0.4618 | −0.4618 | −0.4618 | −0.4618 | −0.4618 | −0.4618 | −0.4618 | −0.4618 | −0.4618 | −0.4618 | 2.0012 | 2.0012 | 2.0012 |
| $\dot{a}_{41}^{T_n}$ | −1.1126 | −1.1126 | −1.1126 | −1.1126 | −1.1126 | −1.1126 | −0.6111 | 0.1410 | 0.8932 | 0.8932 | 0.8932 | 0.8932 | 0.8932 | 0.8932 | 0.8932 | 0.8932 |
| $\dot{a}_{42}^{T_n}$ | −2.5649 | −0.5130 | −0.5130 | −0.5130 | −0.5130 | −0.5130 | −0.5130 | −0.5130 | 0.5130 | 0.5130 | 0.5130 | 0.5130 | 0.5130 | 0.5130 | 1.5390 | 1.5390 |
| $\dot{a}_{43}^{T_n}$ | −0.5533 | −0.5533 | −0.5533 | −0.5533 | −0.5533 | −0.5533 | −0.5533 | −0.5533 | −0.5533 | −0.5533 | −0.5533 | −0.5533 | 1.6599 | 1.6599 | 1.6599 | 1.6599 |

**Table A3.** Order degree of indicator parameters.

| | T1 | T2 | T3 | T4 | T5 | T6 | T7 | T8 | T9 | T10 | T11 | T12 | T13 | T14 | T15 | T16 |
|---|---|---|---|---|---|---|---|---|---|---|---|---|---|---|---|---|
| $U_{T_n}(a_{11})$ | 0.0000 | 1.0000 | 1.0000 | 1.0000 | 1.0000 | 1.0000 | 1.0000 | 1.0000 | 1.0000 | 1.0000 | 1.0000 | 1.0000 | 1.0000 | 1.0000 | 1.0000 | 1.0000 |
| $U_{T_n}(a_{12})$ | 0.3333 | 0.3333 | 0.1667 | 0.0000 | 0.6667 | 0.6667 | 0.6667 | 0.3333 | 0.5000 | 0.5000 | 0.5000 | 0.5000 | 1.0000 | 1.0000 | 1.0000 | 1.0000 |
| $U_{T_n}(a_{13})$ | 0.0000 | 0.0000 | 0.0000 | 0.0000 | 0.3750 | 0.3750 | 0.3750 | 0.3750 | 0.6250 | 0.6250 | 0.6250 | 0.6250 | 1.0000 | 1.0000 | 1.0000 | 1.0000 |
| $U_{T_n}(a_{14})$ | 0.0000 | 0.0000 | 0.0000 | 0.0000 | 0.6154 | 0.6154 | 0.6154 | 0.6154 | 0.3846 | 0.3846 | 0.3846 | 0.3846 | 1.0000 | 1.0000 | 1.0000 | 1.0000 |
| $U_{T_n}(a_{15})$ | 0.0000 | 0.0000 | 0.0000 | 0.0000 | 0.0000 | 0.0000 | 0.0000 | 0.0000 | 0.0000 | 0.7500 | 0.7500 | 0.7500 | 1.0000 | 1.0000 | 1.0000 | 1.0000 |
| $U_{T_n}(a_{21})$ | 0.0000 | 0.0000 | 0.0000 | 0.0000 | 0.0000 | 0.0000 | 1.0000 | 0.0000 | 1.0000 | 1.0000 | 0.0000 | 0.0000 | 1.0000 | 0.0000 | 0.0000 | 1.0000 |
| $U_{T_n}(a_{22})$ | 0.0000 | 0.6667 | 0.0000 | 0.6667 | 0.6667 | 0.6667 | 0.0000 | 0.6667 | 0.6667 | 0.6667 | 0.6667 | 0.6667 | 0.6667 | 0.6667 | 0.6667 | 1.0000 |
| $U_{T_n}(a_{23})$ | 0.0000 | 0.0000 | 1.0000 | 0.0000 | 1.0000 | 1.0000 | 1.0000 | 1.0000 | 1.0000 | 1.0000 | 1.0000 | 1.0000 | 1.0000 | 1.0000 | 1.0000 | 1.0000 |
| $U_{T_n}(a_{24})$ | 0.0000 | 1.0000 | 1.0000 | 0.0000 | 1.0000 | 1.0000 | 1.0000 | 1.0000 | 1.0000 | 1.0000 | 1.0000 | 1.0000 | 1.0000 | 1.0000 | 1.0000 | 1.0000 |
| $U_{T_n}(a_{25})$ | 0.0000 | 0.0000 | 0.0000 | 0.0000 | 0.6667 | 0.6667 | 0.6667 | 0.6667 | 0.6667 | 0.6667 | 0.6667 | 0.6667 | 1.0000 | 1.0000 | 1.0000 | 1.0000 |
| $U_{T_n}(a_{26})$ | 0.0000 | 0.0000 | 0.0000 | 0.0000 | 0.0000 | 0.0000 | 0.3333 | 0.3333 | 1.0000 | 1.0000 | 1.0000 | 0.6667 | 1.0000 | 1.0000 | 1.0000 | 1.0000 |
| $U_{T_n}(a_{27})$ | 0.0000 | 0.0000 | 0.0000 | 0.0000 | 1.0000 | 1.0000 | 1.0000 | 1.0000 | 1.0000 | 1.0000 | 1.0000 | 1.0000 | 1.0000 | 1.0000 | 1.0000 | 1.0000 |
| $U_{T_n}(a_{31})$ | 0.0000 | 0.0000 | 0.0000 | 1.0000 | 1.0000 | 1.0000 | 1.0000 | 1.0000 | 1.0000 | 1.0000 | 1.0000 | 1.0000 | 1.0000 | 1.0000 | 1.0000 | 1.0000 |
| $U_{T_n}(a_{32})$ | 0.0000 | 0.0000 | 0.0000 | 0.0000 | 0.4000 | 0.0000 | 0.0000 | 0.0000 | 0.0000 | 0.0000 | 0.0000 | 0.4000 | 1.0000 | 1.0000 | 1.0000 | 0.0000 |
| $U_{T_n}(a_{33})$ | 0.0000 | 1.0000 | 0.0000 | 1.0000 | 1.0000 | 1.0000 | 1.0000 | 1.0000 | 1.0000 | 1.0000 | 1.0000 | 1.0000 | 1.0000 | 1.0000 | 1.0000 | 1.0000 |
| $U_{T_n}(a_{34})$ | 0.0000 | 0.4000 | 1.0000 | 1.0000 | 1.0000 | 1.0000 | 1.0000 | 1.0000 | 1.0000 | 1.0000 | 1.0000 | 1.0000 | 1.0000 | 1.0000 | 1.0000 | 1.0000 |
| $U_{T_n}(a_{35})$ | 0.0000 | 0.0000 | 0.0000 | 0.0000 | 0.0000 | 0.0000 | 0.0000 | 0.0000 | 0.0000 | 0.0000 | 0.0000 | 0.0000 | 0.0000 | 1.0000 | 1.0000 | 1.0000 |
| $U_{T_n}(a_{41})$ | 0.0000 | 0.0000 | 0.0000 | 0.0000 | 0.0000 | 0.0000 | 0.2500 | 0.6250 | 1.0000 | 1.0000 | 1.0000 | 1.0000 | 1.0000 | 1.0000 | 1.0000 | 1.0000 |
| $U_{T_n}(a_{42})$ | 0.0000 | 0.5000 | 0.5000 | 0.5000 | 0.5000 | 0.5000 | 0.5000 | 0.5000 | 0.7500 | 0.7500 | 0.7500 | 0.7500 | 0.7500 | 0.7500 | 1.0000 | 1.0000 |
| $U_{T_n}(a_{43})$ | 0.0000 | 0.0000 | 0.0000 | 0.0000 | 0.0000 | 0.0000 | 0.0000 | 0.0000 | 0.0000 | 0.0000 | 0.0000 | 0.0000 | 1.0000 | 1.0000 | 1.0000 | 1.0000 |

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
