# Peer review of "Synergy Degree Evaluation of Container Multimodal Transport System"

_sustainability, doi:10.3390/su12041487_

Round 1

Reviewer 1 Report

The paper faces an interesting theme, i.e. how the sinergy strategy can reduce the carbon foot print of multimodal containers logistics. Nevertheless the paper gas ti be reinforced in its introduction and scientific justification. This reinforcement can come from the impact analysis of sinergy to also other actual problems of Supply Chains, such as the bullwhip (see gurbstrom in ijpe), the use of RFID and other iot technologies to minimize the errors and processing time (see fera et al. in 2017 in ijrft) and uncertainty (see fera et al in 2017 in ijiec and uscm).

Author Response

Thank you very much for your suggestions, which made me think more carefully about the argument part of this article. After reading several articles you recommended, I have a better understanding of the uncertainty and coordination issues in the supply chain, which is much helpful for us to revise this manuscript.  Revised on lines 76 to 84.

Reviewer 2 Report

The authors of this paper have done an extensive and impressive research in the fields of social science and data modeling. The background theory and corresponding literature reviews were clearly introduced. The details about data acquisition were thoroughly described. The model structure was sufficiently illustrated. The model results were explicitly presented. The only (relative) weak link is the interpretation on the meaning of results in reality. What do these results mean to decision makers and what benefits could this research offer to policy-making? The authors mentioned a little bit on the conclusions but more elaborations are desired on the significance and meanings of this study to real life transport system, especially on transport policies.

I'd recommend to accept this paper with a few minor revisions as listed below.

line 62
please provide reference for "a survey showed..."

line 361
please elaborate on the entropy weight method

line 483

please elaborate on the potential impacts to policy making based on the modeling results
